# Cancer and Traditional Plant Knowledge, an Interesting Field to Explore: Data from the Catalan Linguistic Area

**DOI:** 10.3390/molecules27134070

**Published:** 2022-06-24

**Authors:** Airy Gras, Montse Parada, Jaume Pellicer, Joan Vallès, Teresa Garnatje

**Affiliations:** 1Laboratori de Botànica—Unitat Associada CSIC, Facultat de Farmàcia i Ciències de l’Alimentació—Institut de Recerca de la Biodiversitat IRBio, Universitat de Barcelona (UB), 08028 Barcelona, Catalonia, Spain; montse.parada@gmail.com (M.P.); joanvalles@ub.edu (J.V.); 2Center for the Study of Human Health, Emory University, Atlanta, GA 30033-5305, USA; 3Institut Botànic de Barcelona (IBB), CSIC-Ajuntament de Barcelona, 08038 Barcelona, Catalonia, Spain; jaume.pellicer@ibb.csic.es (J.P.); tgarnatje@ibb.csic.es (T.G.); 4Royal Botanic Gardens, Kew, Richmond TW9 3AE, UK; 5Secció de Ciències Biològiques, Institut d’Estudis Catalans, 08001 Barcelona, Catalonia, Spain

**Keywords:** antitumor, cancer, cytotoxic activity, ethnobotany, medicinal plants, pharmacological activity, traditional plant knowledge

## Abstract

Cancer is the second cause of death in the world and is foreseen to be responsible for about 16 million deaths in 2040. Approximately, 60% of the drugs used to treat cancer are of natural origin. Besides the extensive use of some of these drugs in therapies, such as those derived from the genus *Taxus*, a significant number of plants have revealed themselves as useful against cancer in recent years. The field of ethnobotany focuses on documenting traditional knowledge associated with plants, constituting a starting point to uncover the potential of new plant-based drugs to treat or prevent, in this case, tumour diseases and side effects of chemotherapy and radiotherapy. From a series of extensive ethnobotanical prospections across the Catalan linguistic area (CLA), we have recorded uses for 41 taxa with antitumour effects. The two most quoted botanical families are Asteraceae and Ranunculaceae, and the most frequently reported species is *Ranunculus parnassifolius*, a high-mountain species, which is widely collected for this purpose. The reported species have been used to treat an important number of cancer types, focusing on preventive, palliative, and curative uses, as well as to deal with the side effects of conventional treatments. Comparing our results in CLA with previous data available in the most comprehensive databases of pharmacology and a review of cytotoxicity assays revealed that for the several species reported here, there was no previous evidence of traditional uses against cancer. Despite the need for further analyses to experimentally validate the information presented here, combining traditional uses and phylogenetically-informed strategies to phytochemical and pharmacological research would represent new avenues to establish more integrative approaches, hence improving the ability to select new candidate taxa in cancer research.

## 1. Introduction

Cancer is one of the leading causes of death worldwide, accounting for nearly 10 million deaths in 2020 [1]. There is about a 20% risk of developing a cancer in a lifetime and a 10% risk of dying from the disease; this means that one in five persons will suffer from some type of cancer in their lifetime and one in ten will, unfortunately, die from the disease [2]. The International Agency for Research on Cancer estimates an incidence of 30 million people and more than 16 million deaths directly linked to this illness by 2040 [3]. Breast cancer was amongst the most diagnosed cancer types in 2020, followed by lung, colon and rectum, prostate, skin (non-melanoma) and stomach cancer. By far, lung cancer, followed by liver and stomach cancer, represent the most deadly types of cancer [2]. The World Health Organization [1] suggests that between 30% and 50% of cancers could be avoided by reducing exposure to risk factors, especially those associated with sedentary lifestyles (e.g., elevated body mass index, low intake of fruits and vegetables and lack of physical activity), bad habits and addictions (including tobacco and recurrent alcohol consumption), or continued exposure to domestic mutagenic agents (e.g., ultraviolet or ionising radiation and air pollution exposure).

Approximately 60% of the drugs used in therapies against cancer are based on chemicals of natural origin [4]. Indeed, because of their recurrent use throughout history, some of them are already considered classical. These include vegetal alkaloids, such as paclitaxel, extracted from *Taxus brevifolia* Nutt. and related species, vinblastine and vincristine from *Catharanthus roseus* (L.) G.Don or cyclolignans, such as podophyllotoxin, extracted from *Podophyllum hexandrum* Royle and *P. peltatum* L. [5].

Among the c. 350,000 vascular plant species described, around 7% of them have documented traditional medicinal uses [6]. However, the search for new drugs is far from being over, and here, ethnobotany can play a very important role in uncovering both new species and uses, and thus contributing to finding new potential drugs for cancer research.

The field of ethnobotany focuses on documenting traditional knowledge and uses associated with plants, providing baseline information for plant screenings to be used during the treatment and/or prevention of certain diseases [7]. Among many other applications, access to such cultural data repositories constitutes a complementary, yet necessary step to further the field of medical research, especially for cancer treatment [8,9]. In fact, the value of ethnobotany in drug research is regulated and even recognised at the legal level. According to the European Medicines Agency (EMA), registration of traditional herbal medicines requires providing a bibliographic track record, or expert’s evidence, regarding the use of a given product as medicinal for a minimum of 30 years prior to the registration process, of which at least 15 must be in the European Union [10]. The ethnopharmacological approach involves several methodologies from social and natural disciplines, and has been for long time the traditional way of selecting plant candidates for drug development [11,12]. However, in recent times, efforts are being made to move the field forward by combining sources of traditional knowledge with more modern “omic” approaches [13,14]. Certainly, the use of molecular phylogenetic frameworks provides unparalleled opportunities for tracing chemical activity and making predictions over evolutionary scales [15,16,17], thus making the identification of a candidate species with potential chemical activity more powerful than ever before (e.g., antineoplastic activity) [18].

Phytotherapy and pharmaceutical ethnobotany have been frequently used to address mild and chronic illnesses, but nonetheless, they have also proved to be useful against acute and more severe health issues, such as cancer [19,20]. Interest in medical plant research is not new, as back in 1994, The Lancet editorial [21] emphasised the need for investing more funds in the study of plants as the basis for new drug discovery, referring, among others, to cancer treatment. Nowadays, unlike in the past, natural medicinal product research considers the ecological consequences of overharvesting plants from the wild and the impact of exploiting biodiversity. Likewise, international agreements, legislations and conservation strategies have been developed to protect both the biodiversity as well as the ethical implications of traditional knowledge of indigenous communities [12,22].

Ethnobotanical studies focusing on plant applications for cancer research have been published worldwide [23,24,25,26,27,28,29], but never before has a study been published focusing on the Iberian Peninsula, nor the Catalan linguistic area (CLA). Based on this, and aiming at filling such caveat, the main objectives of this study were as follows: (i) to report the potential uses of plants in cancer treatment, including preventive and palliative applications, based on the information of traditional uses recorded in CLA; (ii) to carry out a literature review of pharmacological activity and uses of plants concerned; and (iii) to prove the importance of traditional knowledge as a starting point in the development of new plant-based drugs.

## 2. Results and Discussion

### 2.1. General Data

Over the last 30 years of ethnobotanical prospection across the Catalan linguistic area, we have gathered information on folk plant uses related to cancer for 41 plant species, including references to curative, palliative or preventative purposes (Table 1). It is however, worth mentioning that in some cases, the informants use local euphemisms to describe the origin of the illness, such as “mal dolent” or “mal lleig”, which literally translate to bad or ugly illness [30]. One explanation for this is that still today, it is hard for many people to use the word cancer, being considered to some extent a taboo disease (or forbidden word). This is especially common among the elderly, and is clearly associated with traditional knowledge. Certainly, fears behind the use of the word cancer could be most likely associated with the high rates of deaths attributed to this illness, prior to the improvement of detection and subsequent therapies of treatment.

A systematic review of the traditional uses of medicinal plants to treat cancer was published in 2021 [31]. The review reported a total of 948 plants species used against cancer around the world. Surprisingly, despite the large number of species reported in this former global review, 21 (out of the 41) taxa reported in our study were not included before in any compilation of uses, and also, a total of 37 taxa are quoted for the first time for the Iberian Peninsula and the Balearic Islands following this review publication. Altogether, these findings contribute significantly to improve our understanding of traditional plant uses related to cancer management, but also emphasise the power of in-depth surveys at regional and local levels to uncover new potential candidate species, and thus complement global initiatives in cancer research.

### 2.2. Most Recurrent Taxa Used by Informants in Prospected Area

The most cited plant against cancer in our surveys across the area prospected was *Ranunculus parnassifolius* (Figure 1), a perennial herb belonging to the Ranunculaceae family, a botanical family widely studied over centuries in traditional ethnomedicine. In particular, this species grows in high-mountain screes in the Pyrenees and in other European mountain ranges. In the Catalan Pyrenees, this taxon is traditionally used for the treatment of “mal gra”, a Catalan term that translates into “bad pimple”, frequently used to designate some sort of skin cancer. It is usually prepared by combining it with chicken fat in the form of balm or unguent [35]. This plant is popularly very strongly associated with this specific use that it is, indeed, widely known by its local Catalan name “herba del mal gra” (i.e., “bad pimple herb”) [30]. One of the negative consequences of this medicinal reputation is that, in recent years, this taxon has been intensively collected in the wild. Besides the impact of climate change in high mountain ecosystems, uncontrolled harvesting adds an extra level of threat to already damaged populations, many of which suffer the consequences of uncontrolled human activities, such as tourism and high-mountain sports, in these areas. Certainly, as already stated by Brower [36], unless action is taken, the rapid loss of biodiversity (to which we add the decline in traditional knowledge, particularly in heavily-industrialised societies) can adversely affect future cancer plant-based drug discovery.

Besides *R. parnassifolius*, another four species belonging to the buttercup family (Ranunculaceae) were also recorded in our study (see Table 1 for details). This result is not surprising, based on the previous investigation by Hao et al. [37], who already pointed out that several phylogenetically-related genera within the family from China contain a series of phytometabolites (e.g., alkaloids, terpenoids, saponins and polysaccharides) with anti-cancer activity [38]. Altogether, this makes the species from the Ranunculaceae family reported here (and in particular *R. parnassifolius*) good potential candidates for future phytochemical and pharmacological investigation.

Another family that was recurrently cited among the informants is the sunflower family (i.e., Asteraceae). According to our results, five species were claimed to be used against cancer by locals (Table 1). One of them is *Silybum marianum*, a very well-studied plant [39,40], which is indeed a great example to illustrate the importance of common plants for cancer research in particular, and for medicine in general. More specifically, *S. marianum* is known for its hepatoprotective properties, which has revealed promising results for cancer treatment in recent research [40]. In addition, other Asteraceae members were reported here with well-established traditional uses (Table 1). With such precedents, future analytical work should then focus on this group of candidate taxa for additional chemical exploration, and thus confirm whether any of these is of anticancer usefulness beyond the current knowledge. Among Rosaceae, four species in the family (Table 1) also play an important role in traditional medicine. Surprisingly, the utilised part of these species in the studied territories is not the fruit. In general, the presence of phytochemicals and antioxidants in Rosaceae fruits and their potential as cancer inhibitors are well known [41], but based on our results, investigating alternative tissues could indeed open new avenues in this field of research.

Beyond the above-mentioned botanical families, other plant species were reported to fight against the more frequent cancers associated with high mortality rates in rural areas. Among them, *Helleborus foetidus*, *Plantago lanceolata* and *Plantago major* have been reported by our informants to treat skin carcinogenic injuries, *Thymus vulgaris* for throat-related forms of cancer, *Crocus sativus* for breast tumors, *Anemone hepatica* for liver, and *Malva sylvestris* for colorectal cancers (Table 1).

### 2.3. Wild and Cultivated Vegetables and Their Role against Cancer

In the area of preventive strategies against cancer by WHO [1], specialists pointed out the low consumption of fruit and vegetables as an important cancer risk factor. Even if the relationship between food and vegetable intake and cancer does not seem to be clear in most cases, their consumption is encouraged in trouble-preventive healthy diets [42,43]. In this respect, we want to emphasise the idea around the role of folk functional foods or nutraceuticals [44] as preventative and curative. For example, the species *Brassica oleracea* (wild cabbage and its common infraspecific categories including broccoli, cauliflower, kale, etc.), was widely reported as an effective source against stomach cancer by our informants. In addition, epidemiological studies highlight the positive effects of the ingestion of plants belonging to the genus *Brassica* as a cancer preventive [45], providing support to our reports. In addition, four frequently cultivated vegetables (*Allium cepa* and *Daucus carota* subsp. *sativus* were specifically cited for the treatment of stomach cancer; *Apium graveolens* and *Beta vulgaris*), a minor or neglected crop (*Helianthus tuberosus*), and a wild plant often consumed as a vegetable (*Urtica* sp.) complete the set of folk functional foods potentially useful as antitumour sources (representing 17.5% of the cited species; Table 1). Altogether, these results show the importance of plants (both cultivated and their wild crop relatives), whose regular intake in diets can be beneficial and cancer preventive.

### 2.4. Plants for Dealing with Side Effects of Cancer Treatments

As highlighted in previous sections, this study provides compelling evidence regarding the importance of assessing traditional knowledge linked to plants as a possible source of new drugs. Whilst efforts need to focus on the prevention and/or cure of cancer, another important issue to tackle is the side effects of standard medical treatments, and investigation into how plants can contribute to the alleviation of the effects of antitumour therapies. Reducing (or minimising) the negative effects—e.g., anaemia, appetite loss, nausea and vomiting, general pain and distress, mouth infections, among others—of chemotherapy and radiotherapy treatments is one of the goals that is being pursued in this field, and plants are also present in this line of research [46]. Based on our research, the most common species reported with analgesic activity (i.e., pain relief) are *Cannabis sativa* and *Papaver somniferum*, whilst *Plantago sempervirens* and *Santolina chamaecyparissus* were used for mouth infections derived from therapies against cancer.

### 2.5. Pharmacological Activity Review

The official monographs of the European Medicines Agency (EMA) [32] and European Scientific Cooperative on Phytotherapy (ESCOP) [47] include just one out of the forty-one species reported in our study. According to the EMA records, the monograph report for *Allium cepa* provides details regarding the anticarcinogenic and antimutagenic activities associated with this species [32]. Consulting other sources of literature on phytotherapy made it possible to validate a total of 68.29% of the species reported here. In fact, Duke’s CRC *Handbook of Medicinal Herbs* [33] was by far the most inclusive, systematic and detailed work analysed of any of the sources consulted (Table 1).

In addition, we carried out a review of the cytotoxicity tests against cancer cell lines for the plants reported by informants in our field surveys. So far, a total of 26 species have been the focus of different studies to test inhibition of cell growth in cancer cell lines (see Table 2 for details). Some of the species currently lacking any information regarding the inhibitory activity, such as *Plantago major*, *Ranunculus bulbosus* or *Ranunculs parnassifolius,* have been otherwise well studied from a genus level perspective, and the results obtained in related species could be similar for these taxa, although this would require future confirmation. In fact, the lack of cytotoxicity assays does not necessarily mean that some of these species are not indeed active against cancer. Other pharmacological activities, such as antioxidant capacity by scavenging reactive oxygen, are important in preventing potential damage to cellular components such as DNA, proteins and lipids. The oxidative damage can cause major problems, such as carcinogenesis [48]. Some of the species reported here are not studied against cancer cell lines such as *Apium graveolens*, *Rubus ulmifolius* or *Sambucus nigra*, but are well known for their antioxidant activity [49,50,51], and could be, therefore, good candidates to test in future assays.

In summary, our study provides relevant information on the traditional uses of plants against cancer across the Catalan linguistic area, contributing to the global understanding of ethnomedicine to mitigate the impact of an illness that kills nearly 10 million people per year. We want to stress, however, that our efforts to collate and make such data available should be paralleled by an increase in pharmacological studies to experimentally validate the data reported here, and also by subsequent analyses regarding the impact of these chemicals on cancer cell lines.

## 3. Materials and Methods

### 3.1. Studied Area

The Catalan linguistic area constitutes a well-studied area from the following several perspectives: geographic [97], physiographic [98], floristic [99,100], vegetation [101], linguistic and cultural approach [102]. This territory, located in the eastern part of the Iberian Peninsula, also includes a northern Pyrenean portion, the Balearic Islands, and the city of L’Alguer on the island of Sardinia. Politically, this territory belongs to the following four states: Andorra (all the territory), France (Northern Catalonia or Eastern Pyrenees department), Italy (L’Alguer, Sardinia), and Spain (Balearic Islands, Carxe—a small area in Murcia, Catalonia, a portion of eastern Aragon, and Valencia) (Figure 2). It is home to around 14,000,000 people [103,104,105,106,107] and extends across 70,000 km^2^ [100].

The orographic profile is quite diverse, from the Mediterranean Sea level to 3143 m a.s.l. at the summit of Pica d’Estats (Pyrenees). The landscape of the area of study is structured in several belts with distinct floristic and vegetation traits [99,100], harboring approximately 4300 autochthonous and 1200 allochthonous plant taxa, including species and subspecies [108].

### 3.2. Databasing and Data Selection

The information was collected through semi-structured ethnobotanical interviews [109], following the ethical principles of the International Society of Ethnobiology [110], and included in an open-access webpage (https://etnobotanica.iec.cat), which contains the ethnobotanical data for the Catalan linguistic area [111]. Herbarium vouchers were prepared for each species and are deposited in the herbarium BCN (Centre de Documentació de Biodiversitat Vegetal, Universitat de Barcelona). All the information available concerning plants used to treat, palliate or prevent cancer was retrieved from the open access webpage mentioned before. Data obtained from informants were compiled from interviews performed from 1990 to the present.

Bolòs et al.’s [100] was followed for taxonomic nomenclature, which is a specifically flora focused on the studied area. Plants of the World Online (https://powo.science.kew.org) was also consulted when exotic plants were involved. For family attribution, we followed the criteria stablished by the APG IV, the last Angiosperm Phylogeny Group’s arrangement to date [112].

### 3.3. Pharmacological Activity Review

In order to confidently assess how many of the species cited in our surveys (i.e., the Catalan linguistic area) had been previously studied in depth, a review of pharmacological activities was carried out. Monographs of the official sources, such as the European Medicines Agency (EMA) [32], the European Scientific Cooperative on Phytotherapy (ESCOP) [47], the encyclopedic bibliography on phytotherapy as Duke’s CRC “*Handbook of Medicinal Herbs*” [33] and Fitoterapia.net webpage [34], were consulted.

In addition, an original publication search was carried out using four major online databases for scientific bibliographic resources, namely PubMed, ScienceDirect, Scopus and Web of Science, using the following set of keywords: “scientific name” AND “cancer” AND “chemical compounds”. The aim of this search was mainly to review the existence of cytotoxic activity tests in cancer cell lines involving the plant species of interest.

## 4. Conclusions

Ethnobotanical research, when integrated into new fields of study, such as molecular phylogenetics, phytochemistry and other “omic” disciplines, can become a powerful tool to experimentally validate traditional plant knowledge, as well as to predict promising new sources of plant-based drugs. This study represents a step forward in our understanding of folk medicine as a resource to obtain relevant information to treat symptoms related to cancer. We are, nonetheless, aware that there is still a long way to go to before this information can be used in oncological procedures. Plants with an ethnobotanical tradition deserve to be studied more thoroughly, as they provide potential candidate scenarios to fight against cancer and its associated side effects. It is, of course, important to continue to stay in alliance with conventional medicine, which already includes plant-based products. Based on this, ethnobotanical research should become a standard tool in pharmacological and medicinal research, as it could help to guide pathways for drug discovery, and play a significant factor when applying new solutions to the many rising challenges when fighting against diseases such as cancer.

## Figures and Tables

**Figure 1 molecules-27-04070-f001:**
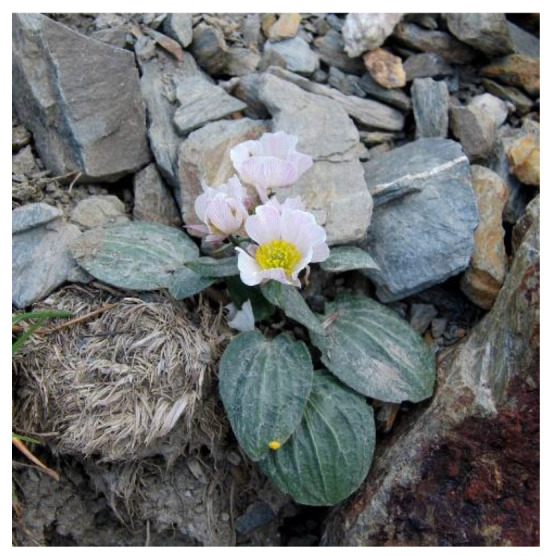
*Ranunculus parnassifolius* in its rocky high mountain habitat in the Catalan Pyrenees (image photo: Albert Mallol Camprubí).

**Figure 2 molecules-27-04070-f002:**
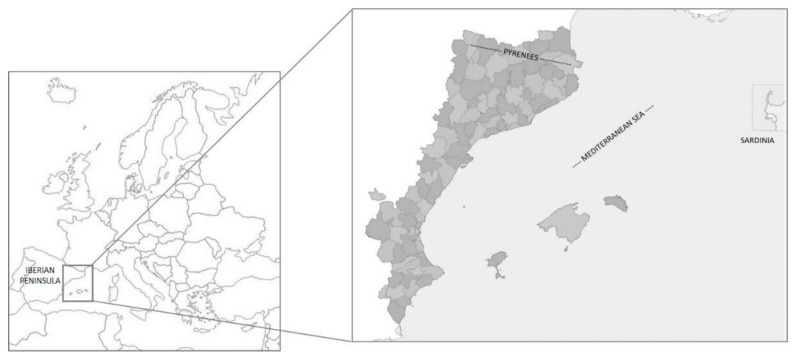
Map of Catalan linguistic area within Europe.

**Table 1 molecules-27-04070-t001:** Plant taxa used against cancer in the Catalan linguistic area, grouped by curative, palliative, and preventive activities. Comparison of uses in the pharmacological comprehensive literature was extracted from: European Medicines Agency (EMA) monographs [32], Duke’s CRC *Handbook of Medicinal Herbs* [33], and Fitoterapia.net webpage [34].

Taxon (Herbarium Voucher)	Family	Plant Part Used	Pharmaceutical Form	Pharmacological Literature
**Curative**				
*Abies alba* Mill. (BCN 24699)	Pinaceae	Resin	Without pharmaceutical form (internal use)	[33]
*Agrimonia eupatoria* L. (BCN 24704)	Rosaceae	Aerial part	Unknown	[33]
*Allium cepa* L. (BCN 27279)	Amaryllidaceae	Bulb	Tisane	[32,33,34]
*Anemone hepatica* L. (BCN 27247)	Ranunculaceae	Leaf	Tisane	
*Angelica sylvestris* L. (BCN 24712)	Apiaceae	Unknown	Unknown	
*Brassica oleracea* L. (BCN 24728)	Brassicaceae	Leaf	Tisane/Without pharmaceutical form (internal use)	[33,34]
*Bryonia cretica* L. (BCN 24730)	Cucurbitaceae	Root	Without pharmaceutical form (topical use)	[33]
*Clematis flammula* L. (BCN 29856)	Ranunculaceae	Aerial part	Poultice	
*Crocus sativus* L. (BCN 32170)	Iridaceae	Styles and stigmas	Poultice	[33,34]
*Daucus carota* L. subsp. *sativus* (Hoffm.) Arcang. (BCN 46847)	Apiaceae	Root	Tisane	[33,34]
*Ecballium elaterium* (L.) A.Rich (BCN 46846)	Cucurbitaceae	Aerial part	Unknown	
*Geranium robertianum* L. (BCN 24894)	Geraniaceae	Aerial part	Tisane	[33]
*Helleborus foetidus* L. (BCN 29705)	Ranunculaceae	Aerial part	Poultice	
*Malva sylvestris* L. (BCN 24924)	Malvaceae	Aerial part/Flower	Tisane	[33]
*Plantago lanceolata* L. (BCN 24949)	Plantaginaceae	Leaf	Without pharmaceutical form (topical use)	[33]
*Plantago major* L. (BCN 24950)	Plantaginaceae	Aerial part/Leaf	Without pharmaceutical form (internal and topical use)	[33]
*Potentilla reptans* L. (BCN 47660)	Rosaceae	Leaf	Tisane	[33]
*Prunella vulgaris* L. (BCN 29759)	Lamiaceae	Flower	Tisane	[33]
*Prunus dulcis* (Mill.) Weeb. (BCN 46833)	Rosaceae	Resin	Without pharmaceutical form (topical use)	[33]
*Ranunculus bulbosus* L. (BCN 24966)	Ranunculaceae	Root	Without pharmaceutical form (topical use)	[33]
*Ranunculus parnassifolius* L. (BCN 24967)	Ranunculaceae	Root/Whole plant	Tisane/Without pharmaceutical form (topical use)	
*Rubus ulmifolius* Schott (BCN 24978)	Rosaceae	Tender bud	Tisane	
*Ruta chalepensis* L. (BCN 24980)	Rutaceae	Aerial part	Unknown	[33]
*Scrophularia alpestris* J.Gay ex Benth. (BCN 29790)	Scrophulariaceae	Young leaf	Without pharmaceutical form (topical use)	
*Silybum marianum* (L.) Gaertn. (BCN 29958)	Asteraceae	Inflorescence	Unknown	[33,34]
*Tetragonia tetragonioides* (Pall.) Kuntze (BCN 29805)	Aizoaceae	Young aerial part	Without pharmaceutical form (internal use)	
*Thymus vulgaris* L. (BCN 25023)	Lamiaceae	Flowering aerial part	Gargarism	[33]
*Verbena officinalis* L. (BCN 25036)	Verbenaceae	Aerial part	Poultice	[33]
*Viola sylvestris* Lam. (BCN 26791)	Violaceae	Leaf	Tisane	
**Palliative**				
*Cannabis sativa* L. (BCN 24735)	Cannabaceae	Leaf/Young aerial part	Tisane/Poultice	[33,34]
*Papaver somniferum* L. (BCN 24941)	Papaveraceae	Latex	Unknown	[33]
*Plantago sempervirens* Crantz (BCN 96761)	Plantaginaceae	Flowering aerial part	Mouthwash	
*Santolina chamaecyparissus* L. (BCN 24986)	Asteraceae	Flowering aerial part	Mouthwash	[33]
**Preventive**				
*Achillea millefolium* L. (BCN 24700)	Asteraceae	Inflorescence	Tisane	[33]
*Angelica sylvestris* L. (BCN 24712)	Apiaceae	Unknown	Unknown	
*Apium graveolens* L. (BCN 24714)	Apiaceae	Aerial part	Unknown	[33]
*Beta vulgaris* L. subsp. *vulgaris* var. *conditiva* Alef. (BCN 52089)	Amaranthaceae	Root	Tisane/Without pharmaceutical form (internal use)	[33]
*Brassica oleracea* L. (BCN 24728)	Brassicaceae	Leaf	Without pharmaceutical form (internal use)	[33,34]
*Calendula arvensis* L. (BCN 32863)	Asteraceae	Inflorescence	Liniment	
*Helianthus tuberosus* L.(BCN 24898)	Asteraceae	Tuber	Boiled	
*Petroselinum crispum* (Mill.) Hill (BCN 24943)	Apiaceae	Leaf	Poultice	[33]
*Ranunculus parnassifolius* L. (BCN 24967)	Ranunculaceae	Root/Whole plant	Tisane/Without pharmaceutical form (topical use)	
*Sambucus nigra* L. (BCN 24984)	Adoxaceae	Inflorescence	Fumigation	[33]
*Urtica* sp.	Urticaceae	Unknown	Tisane	[33]

**Table 2 molecules-27-04070-t002:** Review of the cytotoxic activity against cancer cell lines for plants quoted in the Catalan linguistic area to treat cancer.

Taxon	Plant Part Used	Extract	Chemical Compound	Cell Line	Cytotoxic Activity (Key Results)	Reference
*Abies alba* Mill.	Seed and cone	Aq	-	MCF7 and MDA-MBA-231	The influence of the essential oils on the cancer cells was weak. The IC_50_ values were similar to those found towards normal cells (100 µg/mL)	[52]
*Achillea millefolium* L.	-	EtOH	Phenolic acids (3,5-O-dicaffeoylquinic acid, 5-O-caffeoylquinic acid), flavonoids (luteolin-O-acetylhexoside, apigenin-O-acetylhexoside)	NCI-H460 and HCT-15	The extract showed an inhibitory effect on the growth of NCI-H460 and HCT-15 cell lines with IC_50_ values 187.3 µg/mL and 70.8 µg/mL, respectively	[53]
*Agrimonia eupatoria* L.	Aerial part	Aq and MeOH	-	RD and HeLa	The extracts showed anti-tumor properties in a concentration-dependent manner, and the MeOH extract recorded better values of percentage of growth inhibition than aqueous extract in HeLa and RD cell lines (IC_50_: 96 µg/mL)	[54]
*Allium cepa* L.	Bulb	MeOH	-	HeLa, HCT 116 and U2OS	The IC_50_ values obtained were 24.79, 24.73 and 36.6 µg/mL for HeLa, HCT 116 and U2OS cell lines, respectively	[55]
	Bulb	MeOH	Quercetin and quercetin 4′-O-β-glucoside	B16	Quercetin and quercetin 4′-O-β-glucoside compounds showed inhibition in B16 cells with IC_50_ values of 26.5 and 131 µM, respectively	[56]
	Flower	MeOH	Polyphenols	K562, THP-1 and U937	The results revealed IC_50_ value less than 40 µg/mL for U937 cells and 60 µg/mL for THP-1 and K562	[57]
*Beta vulgaris* L. subsp. *vulgaris* var. *conditiva* Alef.	Root	EtOH	-	AGS	The highest concentration of extract (0.05%) induced significantly greater early apoptosis in relation to the other concentrations. At the same time, it activated the lowest level of late apoptosis and necrosis in AGS cells	[58]
*Brassica oleracea* L.	Sprout	Hx	Sulforaphane	AGS and MKN45	Significant dose-dependent and anti-proliferative effects were observed on AGS and MKN45 cells, with an IC_50_ value of about 112 and 125 μg/mL, respectively	[59]
	Leaf	HCl MeOH	-	HeLa and Hep G2	The IC_50_ values of the extract were 23.38 and 28.66 mg/mL for HeLa and Hep G2, respectively	[60]
	-	-	Sulforaphane, iberin and iberverin	A549	The IC_50_ values were 3.53, 4.93 and 7.07 µg/mL for sulforaphane, iberin and iberverin, respectively	[61]
*Bryonia cretica* L.	Root	EtOH	Cucurbitacin B and E	U937	The cucurbitacin B and E showed great effects with IC_50_ values of 9.2 and 16 nM	[62]
	Root	Aq	-	BL41	The IC_50_ of extract was estimated to be approximately 15.63 µg/mL	[63]
*Calendula arvensis* L.	Inflorescence	Aq and MeOH	-	AML	The extracts exhibited activity against AML (IC_50_: 31 mg/mL)	[64]
*Cannabis sativa* L.	-	-	Cannabidiol (1), tetrahydrocannabinol (2) and cannabinol (3) para-quinones	Raji, Jurkat E6-1, SNB-19, MCF7, DU 145, NCI-H-226 and HT-29	The three compounds displayed antiproliferative activity in all cell lines	[65]
	Aerial part	Aq, Hx, DCM, DCM:MeOH and MeOH	-	Caco-2, HCT-15, HT-29, LS513	Aq and DCM:MeOH extracts moderately inhibited the growth in HCT-15 and LS513 cells (IC_50_: 20–100 µg/mL). Aq and DCM extracts potently inhibited HT-29 cell growth (IC50: 7.52–10.06 µg/mL). Hx and DCM extracts slightly stimulated growth in Caco-2 cells (IC50: 100 µg/mL)	[66]
*Clematis flammula* L.	Aerial part	-	-	HCT 116	The extract showed apoptosis in HCT 116 cell lines	[67]
*Crocus sativus* L.	Leaf	PET	Crocetin (β-D-glucosyl) ester	MCF7	The antiproliferative activity of the compound against MCF7 cell line has showed inhibitory effect in a dose-dependent way with IC_50_ value of 628.36 µg/mL	[68]
	Stigma	-	-	HeLa, A-204 and Hep G2	All tested cell lines showed a good response to the effect of the saffron extract (50–400 µg/mL), but the A-204 cells showed a higher sensitivity to the inhibitory effect	[69]
	Stigma	MeOH	-	AGS, MDA-MB-468 and U-87	The IC_50_ are between 0.8 and 4.5 mg/mL	[70]
	Stigma	-	Crocetin	A549, B16-F10, MCF7 and SK-OV-3	The IC_50_ were 79.79, 55.39, 270.13 and 559.0 µg/mL for MCF7, A549, B16-F10 and SK-OV-3, respectively	[71]
	Stigma	-	Phenols	Caco-2	A significant 32% decrease in Caco-2 cell viability was observed, but only at a concentration of 50 µL/mL	[72]
	-	-	Crocin and safranal	K-562	Drug cytotoxicity experiments showed a dose-dependent cell growth inhibition after exposure of cells to crocin and safranal withIC_50_ values of 160.00 μM and 241.00 μM, respectively	[73]
*Daucus carota* L. subsp. *sativus* (Hoffm.) Arcang.	Root	MeOH	6-Methoxymellein	MCF7 and MDA-MB-231	The compound induced suppression of proliferation at >0.8 mM (MDA-MB-231) and >0.5 nM (MCF7)	[74]
*Ecballium elaterium* (L.) A.Rich	Fruit	MeOH	Cucurbitacin D, E and I	AGS	The cytotoxic effects on AGS gastric cancer cell line showed that cucurbitacin E has greater cytotoxicity in comparison with cucurbitacins D and I. The IC_50_ values were 0.3, 0.1, and 0.5 μg/mL for cucurbitacins D, E, and I, respectively.	[75]
	Seed	Hx	-	HT-29 and HT-1080	The extract showed a potent antiproliferative HT-29 AND HT-1080 cell lines and the IC_50_ values were 4.86 µg/mL and 4.16 µg/mL, respectively	[76]
	-	-	Cucurbitacin D	NSCLC-N6	The treatment with cucurbitacin D inhibited NSCLC-N6 proliferation (IC_50_: 2.5 µg/mL)	[77]
*Geranium robertianum* L.	-	Aq and EtOH	-	Hep-2p	The extracts showed cytotoxic effect on Hep-2p cancer cells (6.1–25.39% in the EtOH extracts; 0.9–32.5% in the Aq extracts)	[78]
*Helianthus tuberosus* L.	Flower	Hx	-	HT-29 and HCT 116	Feradiol exhibited a significant growth inhibitory effect against HT-29 and HCT 116 cell lines (IC_50_ values of 3.93 and 6.02 μg/mL, respectively)	[79]
	Leaf	EtOAc	4,15-iso-Atripliciolide tiglate	A549, HeLa and MCF7	The compound exhibited significant activity against MCF7, A549 and HeLa (1.97, 7.79, 9.87 µg/mL, respectively)	[80]
*Helleborus foetidus* L.	Whole plant	MeOH	Bufadienolide glucosides	A549 and HL-60	The isolated compounds were cytotoxic to A549 and HL-60 cells, with the IC_50_ values ranging from 0.019 to 3.0 μM	[81]
*Malva sylvestris* L.	Leaf and flower	MeOH	Phenols	A-375 and B16	This extract showed a cytotoxic effect for B16 and A-375 cells, an antiproliferative activity of 97% and 85% with respect to the control	[82]
*Petroselinum crispum* (Mill.) Hill	Leaf and stem	Hx	Phenols	MCF7	The extract tested at 500 μg/mL showed a percentage inhibition of 48.4%, 25.5% and 49.9% on MCF7, MDA-MB-231 and HT-29 cells, respectively	[83]
*Plantago lanceolata* L.	Aerial part	MeOH	Phenols	HeLa, HT-29 and MCF7	The inhibition of cell growth exerted a stronger effect with IC_50_: 172.3, 142.8, 405.5 and 551.7 µg/mL for HeLa, MCF7, HT-29 and MRC-5 cell lines, respectively	[84]
	Leaf	MeOH	Flavonoids	MCF7 and UACC-62	The extracts showed good values for IC_50_ (47.16 and 50.58 µg/mL for MCF7 and UACC-62, respectively)	[85]
*Plantago major* L.	Leaf	MeOH	Flavonoids	MCF7 and UACC-62	The extracts showed good values of IC_50_ (46.5 µg/mL for MCF7 and UACC-62)	[85]
	Seed	MeOH	Triterpene acids	SiHa and Hep G2	The extract exhibited cytotoxic activity for SiHa and Hep-G2 (IC_50_: 174.42 and 246.38 µg/mL, respectively)	[86]
*Potentilla reptans* L.	Aerial part and rhizome	Aq	-	4T1	IC_50_ values were 280.51 μg/mL for rhizome extract and 310.79 μg/mL for aerial parts extract	[87]
	Aerial part	Aq	-	A549 and MCF7	Extract exhibited cytotoxic activity for A549 and MCF7 cells (IC_50_ < 130 µg/mL)	[88]
*Prunella vulgaris* L.	-	Aq	Polysaccharide–zinc complex	Hep G2	The polysaccharide–zinc complex inhibits the proliferation (98.4% inhibition rate at 500 μg/mL) of Hep G2 cells	[89]
*Prunus dulcis* (Mill.) Weeb.	Seed	Ace	Gallic acid and pyrogallol	MCF7 and MDA-MB-468	For MCF7, both compounds showed cytocidal effect at 10 μg/mL, whereas for MDA-MB-468, both compounds showed cytocidal effect at >20 μg/mL	[90]
*Silybum marianum* (L.) Gaertn.	-	-	Silymarin	HCT 116 and SW480	A HCT 116 cells treated with 50, 100, and 200 μM of silymarin reduced the cell growth by 11%, 22% and 48%, respectively. A SW480 cells treated with 50, 100, and 200 μM of silymarin reduced the cell growth by 13%, 28% and 47%	[91]
	-	-	Silybin	Jurkat E6-1	Silybin increased the reduction in Jurkat E6-1 cells in the concentration range of 50–200 μM	[92]
*Thymus vulgaris* L.	Aerial part	Hx	-	U2OS and PANC-1	The essential oil causes a very strong inhibition (60%) of cell viability in PANC-1 cells, compared to 40% of reduction observed in U2OS cells at 10 μg/mL	[93]
	Leaf	EtOH	-	T47D	The extract inhibited 75% of T47D cells at 200 μg/mL	[94]
	Leaf	CHCl_3_	Polyphenol complex	SH-SY5Y and SK-N-BE(2)-C	The extract showed strong levels of cytotoxicity towards SH-SY5Y and SK-N-BE(2)-C cell lines at the highest tested dose level (125.0 µg/mL)	[95]
*Verbena officinalis* L.	Aerial part	Aq	Diacetyl-phenylethanoids	DHD/K12/PROb and HCT 116	Four diacetyl-phenylethanoid compounds exhibited extremely high antiproliferative activity against HCT 116 and DHD/K12/PROb. The IC_50_ values were similar to vinblastine sulfate (1.28 µg/mL)	[96]

**Extract abbreviations:** Ace (acetone); Aq (aqueous); CHCl_3_ (chloroform); DCM (dichloromethane); EtOAc (ethyl acetate); EtOH (ethanol); HCl MeOH (HCl acidified methanol); MeOH (methanol); Hx (hexane); PET (petroleum ether). **Cell line abbreviations:** 4T1 (mouse breast cancer cells); A-204 (human rhabdomyosarcoma cells); A-375 (human melanoma cells); A549 (human lung cancer cells); AGS (human stomach cancer cells); AML (human acute myeloid leukemia cells); B16 (mouse melanoma cells); B16-F10 (mouse melanoma cells); BL41 (human Burkitt’s lymphoma cells); Caco-2 (human colon cancer cells); DHD/K12/PROb (rat colon cancer cells); DU 145 (human prostate cancer cells); HCT 116 (human colon cancer cells); HCT-15 (human colon cancer cells); HeLa (human cervical cancer cells); Hep-2p (human epidermoid laryngeal cancer cells); Hep G2 (human hepatocellular cancer cells); HL-60 (human leukemia cell); HT-1080 (human fibrosarcoma cells); HT-29 (human colorectal cancer cells); Jurkat E6-1 (human lymphoblast cells); K562 (human leukemic cells); LS513 (human colon cancer cells); MCF7 (human breast cancer cells); MDA-MB-468 (human breast cancer cells); MDA-MBA-231 (human breast cancer cells); MKN45 (human gastric cancer cells); NCI-H-226 (human lung cancer cells); NCI-H460 (human non-small cell lung cancer cells); NSCLC-N6 (human non-small cell lung cancer cells); PANC-1 (human pancreatic cancer cells); Raji (human lymphoblast cells); RD (human rabdomyosarcoma cells); SH-SY5Y (human neuroblastoma cells); SiHa (human cervical cancer cells); SK-N-BE(2)-C (human bone marrow neuroblastoma cells); SK-OV-3 (human ovarian cancer cells); SNB-19 (human glioblastoma cells); SW480 (human colon cancer cells); T47D (human breast cancer cells); THP-1 (human leukemic cells); U2OS (human osteosarcoma cells); U-87 (human glioblastoma cells); U937 (human leukemia cells); UACC-62 (human melanoma cells).

## Data Availability

The dataset analysed for this study are available in the manuscript, further inquiries can be directed to the corresponding authors.

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
