# Peer review of "Cancer and Traditional Plant Knowledge, an Interesting Field to Explore: Data from the Catalan Linguistic Area"

_molecules, 2022, doi:10.3390/molecules27134070_

Round 1

Reviewer 1 Report

The manuscript “Cancer and Traditional Plant Knowledge, An Interesting Field to Explore for New Drug Development: Data from the Catalan Linguistic Area’s Ethnobotany” should be re-submitted as a Review Article, not a Research Article as claimed.

The manuscript should be improved, in particular, English editing/proofreading is strongly recommended to enhance the overall manuscript presentation. Mistakes like “intern use” in Table 1?

The title should be revised, especially the phrase “An Interesting Field to Explore for New Drug Development”. The content is not aligned to the title. Drug discovery or drug development or both?

Content in Introduction is not organised. The aims of the manuscript are unclear. What is the significance of this study compared to the review article published in 2021, apart from some species are not been mentioned before?

Authors did not mention the botanist who authenticate the plants.

Table 1: Please replace the “without form” into an appropriate term.

Table 2:

-For chloroform abbreviation, CHCl3 is commonly used.

-In the column title “Part plant used”, please revise.

-In the column title “Chemical compounds”, essential oil is an extract, not compound.

-Suforaphane or Sulforaphane? Authors should carefully check the content of the Table for accuracy. Mistakes were found.

-Crocetin (beta-D-glucosyl)ester, please use symbol β.

Plant’s genus and species name should be written in italics, eg: Ranunculus parnassifolius, Cannabis sativa and many more. Please revise throughout the manuscript.

Lack of critical discussion throughout the manuscript. What is the focus of the study?

Reference style should follow strictly to the Journal’s requirement. Inconsistencies were found.

Author Response

We thank the reviewer for his comments and suggestions.

The manuscript “Cancer and Traditional Plant Knowledge, An Interesting Field to Explore for New Drug Development: Data from the Catalan Linguistic Area’s Ethnobotany” should be re-submitted as a Review Article, not a Research Article as claimed.

The manuscript presented is an original research conducted by our ethnobotanical research group in the Catalan Linguistic Area. In this manuscript, which is not a review, we recorded all the taxa cited by our informants to treat cancer. In addition, we did a comparison with the ethnobotanical review published on this topic in 2021 by Aumeeruddy & Mahomoodally. In addition, a pharmacological literature comparison and a cytotoxicity assays review for the quoted species against cancer in the CLA were carried out.

The manuscript should be improved, in particular, English editing/proofreading is strongly recommended to enhance the overall manuscript presentation. Mistakes like “intern use” in Table 1?

The revised manuscript has been read and corrected by an English native speaker who is, in addition, a botanist (now quoted in the acknowledgements).

The title should be revised, especially the phrase “An Interesting Field to Explore for New Drug Development”. The content is not aligned to the title. Drug discovery or drug development or both?

Content in Introduction is not organised. The aims of the manuscript are unclear. What is the significance of this study compared to the review article published in 2021, apart from some species are not been mentioned before?

We changed the title of the manuscript: Cancer and Traditional Plant Knowledge, An Interesting Field to Explore: Data from the Catalan Linguistic Area’s Ethnobotany

The main aims of this research were: i) to focus on the potential uses of plants in cancer treatment, including preventive and palliative actions, based on the information provided in the Catalan linguistic area; ii) to carry out a literature-based pharmacological review of plants concerned; and iii) to highlight the importance of traditional knowledge as a first step to new drug development.

In the review published in 2021 the results from the ethnobotanical Catalan Linguistic Area research are not considered by the authors, except for Rigat et al. 2007. In the research presented in this manuscript the focus are the plants quoted in this specific area. In addition, a comparison of published data at the ethnobotanical, pharmacological, or cytotoxic level was done. As shown with these two considerations, our manuscript complements the quoted review and goes further.

Authors did not mention the botanist who authenticate the plants.

All the authors are botanists and the identification was done for the authors of the manuscript. We added this information in the section Authors Contribution “AG, JV and TG did the botanical identification of the cited plants in the manuscript.”

Table 1: Please replace the “without form” into an appropriate term.

We replaced all “without form” for “without pharmaceutical form”.

Table 2:

- For chloroform abbreviation, CHCl3 is commonly used. Thanks, changed

- In the column title “Part plant used”, please revise. Thanks, changed (Plant part used)

- In the column title “Chemical compounds”, essential oil is an extract, not compound. Thanks, changed

- Suforaphane or Sulforaphane? Authors should carefully check the content of the Table for accuracy. Mistakes were found. Thanks, we checked all the table

- Crocetin (beta-D-glucosyl)ester, please use symbol β. Thanks, changed

Plant’s genus and species name should be written in italics, eg: Ranunculus parnassifoliusCannabis sativa and many more. Please revise throughout the manuscript.

This must be an application issue, we reviewed the submitted documents and all scientific names were in italics, as we always do, as botanists that we are.

Lack of critical discussion throughout the manuscript. What is the focus of the study?

The discussion was constructed following the main objectives of the study. The principal aim of this study is report the plants quoted in Catalan Linguistic Area to treat cancer. In addition, to evaluate the ethnobotanical information contributed in this manuscript we compare these taxa with other ethnobotanical, pharmacological, or cytotoxic studies to assess our ethnobotanical contribution and highlight which species have not been previously studied.

Reference style should follow strictly to the Journal’s requirement. Inconsistencies were found.

Thanks, we reviewed all the references following the style of the Journal.

Reviewer 2 Report

The aim of this study was to link basic ethnobotanical research,  which is  relevant for experimentally validating traditional plant knowledge with the predicting promising new sources of oncology drugs. The plant species of interest were those from the Catalan linguistic area.

This research is important and can bring valuable information with implications for future research and antitumoral therapy.

Therefore, the paper is of interest, but some points must be considered prior acceptance.

Overall, the work is well written and organized, although there are some typing errors to correct during the revision of the work and some scientifically data are missing. All Latin words must be written in italics.

The studies taken in the analysis are valuable, relevant and suitable but the chapter 2.5. Pharmacological Activity Review presents only in vitro studies performed on different tumor cell lines, but in vivo studies could be more relevant. It may have been useful to consider also other types of animal or even clinical studies and the results could be presented in a table, and the bibliography should be completed in this case. Only in vitro cell line test results are shown in the table 2. 

In all the text the Latin name of the plant species must be written in italics according to the standards. 

Author Response

The aim of this study was to link basic ethnobotanical research, which is relevant for experimentally validating traditional plant knowledge with the predicting promising new sources of oncology drugs. The plant species of interest were those from the Catalan linguistic area.

This research is important and can bring valuable information with implications for future research and antitumoral therapy. Therefore, the paper is of interest, but some points must be considered prior acceptance.

We thank the reviewer for his comments and suggestions.

Overall, the work is well written and organized, although there are some typing errors to correct during the revision of the work and some scientifically data are missing. All Latin words must be written in italics.

We checked carefully all the manuscript and we correct some typing errors. The error of the Latin names must be an application issue, we reviewed the submitted documents and all scientific names were in italics, as we always do as botanists that we are.

The studies taken in the analysis are valuable, relevant and suitable but the chapter 2.5. Pharmacological Activity Review presents only in vitro studies performed on different tumor cell lines, but in vivo studies could be more relevant. It may have been useful to consider also other types of animal or even clinical studies and the results could be presented in a table, and the bibliography should be completed in this case. Only in vitro cell line test results are shown in the table 2. 

The section 2.5. consider different types of data. On the one side, we did a review of official monographs and literature on phytotherapy and those sources consider in vivo studies. On the other side, we summarized in vitro cytotoxicity assays to assess whether the species mentioned had been studied before in a simple way as a cytotoxicity assay. We are aware that in vivo studies could also be included in this table, but the aim of the manuscript is presenting our data and to check roughly which of the species quoted had been studied or not before.

In all the text the Latin name of the plant species must be written in italics according to the standards. 

This must be an application issue, we reviewed the submitted documents and all scientific names were in italics, as we always do as botanists that we are.

Round 2

Reviewer 1 Report

The resubmitted manuscript “Cancer and Traditional Plant Knowledge, An Interesting Field to Explore: Data from the Catalan Linguistic Area’s Ethnobotany” should be improved.

The revised title has grammatical error. Please revise.

Please read again the whole manuscript for the English language and grammar. Errors are detected.

Table 2: cytotoxicity activity should be cytotoxic activity. Again, please check the Tables for their accuracy and consistency.

Please be consistent in the unit used, ml or mL? µl or µL? Again, please be consistent.

Author Response

The resubmitted manuscript “Cancer and Traditional Plant Knowledge, An Interesting Field to Explore: Data from the Catalan Linguistic Area’s Ethnobotany” should be improved.

The revised title has grammatical error. Please revise.

We changed the title of the manuscript: Cancer and Traditional Plant Knowledge, An Interesting Field to Explore: Data from the Catalan Linguistic Area

Please read again the whole manuscript for the English language and grammar. Errors are detected.

Thanks, our colleague Jaume Pellicer did a critical revision of the manuscript and an extensive English language revision. We added him as an author of the manuscript.

Table 2: cytotoxicity activity should be cytotoxic activity. Again, please check the Tables for their accuracy and consistency.

Thanks, we reviewed again all the table

Please be consistent in the unit used, ml or mL? µl or µL? Again, please be consistent.

We homogenized all the units, mL and µL

Reviewer 2 Report

In accordance with the suggestions, the changes made by the authors   bring clarifications and improve the quality of the manuscript.

Author Response

In accordance with the suggestions, the changes made by the authors bring clarifications and improve the quality of the manuscript.

Thanks